# The Association of Moral Injury and Healthcare Clinicians’ Wellbeing: A Systematic Review

**DOI:** 10.3390/ijerph20136300

**Published:** 2023-07-05

**Authors:** Pari Shah Thibodeau, Aela Nash, Jennifer C. Greenfield, Jennifer L. Bellamy

**Affiliations:** Graduate School of Social Work, University of Denver, 2148 S. High Street, Denver, CO 80210, USA

**Keywords:** moral injury, wellbeing, healthcare workers, healthcare professionals, workforce, systematic review

## Abstract

Research focused on elucidating moral injury amongst healthcare workers (HCWs) is essential due to the deep connection with morality and individuals’ overall wellbeing. Examining moral injury provides an avenue through which researchers can connect individual experiences with systemic level causes (i.e., structural power imbalances between clinicians and health systems) to better study workplace wellbeing. The omnipresence of the COVID-19 pandemic has amplified the need to study moral injury. This paper describes a systematic review conducted using PRISMA-P guidelines to answer the question, “what is the association between moral injury and professional wellbeing and mental health amongst healthcare workers.” Twelve databases were searched to identify specified studies. This study’s criteria included: (1) articles published through December 2022; (2) qualitative and quantitative empirical studies; (3) articles written in English; (4) articles including moral injury; and (5) articles including at minimum one other measure of professional or personal wellbeing. The initial search produced 248 articles, and 18 articles were ultimately included in the final review. To confirm that no articles were left out of this study, the first author of each included article was contacted to inquire about any additional works that met the inclusion criteria of this study. The elements of the 18 included articles described in this review are discussed. The results indicate that moral injury is associated with both professional wellbeing factors and mental health outcomes. Further theoretical development, including (professional- and identity-based) exploratory research on moral injury, and evidenced-based interventions for moral injury are needed.

## 1. Introduction

In the United States, there are 22 million people working within the healthcare system [1]. In a recent report, “Clinicians of the Future”, only 57% of healthcare workers (HCWs) believed that they have a good work–life balance [2]. One reason why achieving a good work–life balance in healthcare is challenging is due to the high strain the healthcare system in the United States places on HCWs. There have been many indications of this high strain on HCWs such as experiences of burnout, adverse mental health, moral injury, and high turnover [3,4,5]. HCWs experience burnout at a higher rate than other professions [4], and they also experience higher rates of depression, anxiety, post-traumatic stress disorder (PTSD), or suicide than the general population [6]. About 47% of U.S.-based clinicians out of nearly 3000 people surveyed in a recent study stated that they intended to leave their job within the next two to three years [2]. The stress on healthcare workers exacerbates fractures in the services received by patients. For example, when a clinician experiences adverse job-related wellbeing, they are more likely to make a mistake in their work with patients [4]; whether it is a surgeon making a mistake in the operating room or a social worker making a mistake in creating a safe discharge plan for a patient. Declining wellbeing in healthcare workers is risky and harmful to the quality of care provided in hospitals and to patient related outcomes during and post-hospitalization [7].

It is well known that HCWs are not doing well and are in search of further support in their roles [8,9]. The COVID-19 global pandemic further strained U.S. healthcare workers by adding to the pressure of their roles through expectations of heroism and sacrifice, demonstrated by amplified rates of patient abuse towards HCWs [10]. Healthcare work in the pandemic amplified the focus on morality via the question of what “doing the right thing” means. Moral injury was not often discussed in pre-pandemic healthcare discourse; yet, there became an urgent need to better understand the moral and ethical impacts on healthcare workers.

Healthcare workers and policy are present in all systems. Well before COVID-19, schools had nurses, colleges had health and counseling centers, grocery stores had pharmacies, and all of us, when unwell, craved an understanding of how to heal or feel better. Healthcare systems cease to exist without healthcare workers; thus, it is imperative to identify the grave risk of losing healthcare access if our healthcare workers are not being supported as individuals, but rather are treated inhumanely by both the systems and the patients. When addressing human needs regarding wellbeing in high stakes situations such as healthcare, the inclusion of morality is important. A holistic understanding HCWs’ wellbeing can be achieved through studying how moral injury, or harm to individuals’ sense of morality, is associated with wellbeing outcomes amongst the healthcare workforce.

### 1.1. Moral Injury as a Measure of Wellbeing

Moral injury is operationalized in several different ways across the literature. There are two primary definitions of moral injury that are used in the field of ethics and morality. First, Shay’s work was the original definition of moral injury rooted in military populations, which states, “(a) a betrayal of ‘what’s right’; (b) either by a person in legitimate authority or by oneself; (c) in a high stake’s situation” (p. 182) [11]. Shay’s definition is strong due to its explicitness and is simultaneously critiqued for being too rigid. Litz and Kerig’s definition, which is more flexible and applicable to a variety of settings and events, describes moral injury as, “transgressive harms and the outcomes of those experiences’” [12] (p. 341). For this review, a joint definition is used; moral injury is defined as the moral transgression (or boundary breaking) by oneself or someone in a position of power in high stakes situations and the outcomes of those experiences [11,12]. While a range of moral outcomes can be experienced [13] from a potentially morally injurious event (PMIE), a situation in which ethical dilemmas could lead to a moral transgression [12], moral injury is a nuanced experience in which a moral transgression actually occurs. Thus, the experience of moral injury is a unique risk factor to additional adverse wellbeing outcomes. The continuum below identifies and differentiates between the various moral reactions one may have while working in healthcare, ranging from moral frustration to moral distress to moral injury (see Figure 1 below). While there are various moral reactions a HCW can experience, this review specifically focuses on the experience of moral injury; thus, separation of terms is necessary. Moral distress, a construct developed by the nursing field, speaks to the experience of internal or external constraints challenging HCW’s judgements on “what’s right” [13,14]. Moral distress can be a prolonged experience, one which does not always result in completing a moral transgression [14,15]. Moral frustration speaks to the emotional reaction of facing a PMIE [14], and it is the experiences of emotions related to moral challenges that may or may not be directly related to oneself. Figure 1 (below) locates these phenomena across a continuum.

A few studies have highlighted the high prevalence of moral injury amongst healthcare workers prior to and during the COVID-19 pandemic [16,17,18]. During COVID-19, many HCWs experienced moral injury due to shortages of triage, PPE, medication, and supplies [18], highlighting the essential role of adequate resources in healthcare worker wellbeing. In another study, the concept of moral injury resonated with medical students who had observed situations contrary to their own values as future medical professionals and felt that they could not live up to the standards that were required of them [19]. Experiencing moral injury can result in feelings of guilt, shame, and internal confusion with oneself [11,12]. In general, experiencing feelings of guilt and shame is related to overall adverse mental health and wellbeing [20,21]. A further synthesis of how moral injury is then related to professional and personal wellbeing is needed.

### 1.2. Wellbeing Indicators for Healthcare Workers

Workforce wellbeing has been measured using numerous indices and labels contextualizing the high stakes roles of working in healthcare. For this systematic review, HCWs include all patient-facing professions, such as physicians, nurses, social workers, care aids, psychiatrists, psychologists, pharmacists, and physical, occupational, and speech therapists. Generally, the workforce wellbeing literature is divided into two categories: personal wellbeing and professional wellbeing. In this review, professional wellbeing is operationalized as work-related adverse experiences such as burnout, imposter syndrome, compassion fatigue, and turnover. Meanwhile, personal wellbeing refers to the individualized outcomes that impact how one is doing broadly, such as mental health (depression, anxiety, and PTSD) and stress.

Many studies of HCWs professional wellbeing describe high levels of burnout, compassion fatigue, and secondary trauma amongst healthcare clinicians (referring predominantly to doctors and nurses) [22]. Healthcare environments are fast-paced, stressful, and high stakes spaces, and the nature of healthcare has led to many adverse professional wellbeing consequences on HCWs. HCWs’ roles are usually high-pressure and under-resourced, often leading to these negative professional wellbeing outcomes [23,24,25,26,27]. Prior to the COVID-19 pandemic, HCWs were already experiencing high rates of negative professional wellbeing [27], and the global pandemic has only exacerbated these experiences and consequences [28,29,30]. Specifically, across several cross-sectional studies HCWs experienced burnout and exhaustion at the rate of 76% and compassion fatigue at the rate of 52% (*n* = 1119) [31].

Further, HCW wellbeing research has illustrated the impact of healthcare work on individuals’ personal wellbeing, primarily referring to mental health. HCWs experience higher levels of mental health diagnoses and related symptoms [32]. For example, physicians are twice as likely than the general population to die by suicide [33]. Poor wellbeing and high levels of burnout have been associated with poor patient safety outcomes such as medical errors [34]. Moreover, HCWs experience high levels of depression, anxiety, and PTSD related to their jobs [23,28,30]. According to Mental Health America’s 2020 survey, 39% of healthcare workers did not feel adequately emotionally supported—with nurses even less likely to feel supported (45%) [31]. Environments with high exposure to stress and trauma, like healthcare, put HCWs at higher risk of experiencing negative personal wellbeing outcomes. Additionally, personal wellbeing includes spirituality and religiosity. How individuals make meaning of the world can often be derived from spirituality. Moral injury is a concept that is rooted in spirituality and religiosity, and hence the measure of religiosity/spirituality is included in research on moral injury [11].

### 1.3. Positionality Statement

Prior to reading the methods and findings from this review, it is essential for the authors and coders of this work to identify ourselves in relation to the work. The lead author and primary coder is an educated Indian American, cis-gender woman, first-generation college student, and second-generation immigrant who comes from a family that has struggled to attain financial security. Her positionality is essential to name in the present work, as she has experienced moral injury as a healthcare clinical social worker. The second author and secondary coder on this review is a white, queer-identifying, first generation college graduate. She is the child of actors, an older sibling, and comes from a family history of low-economic status, addiction, and untreated mental illness. The third author on this paper is a white, queer, cis-gender woman with experience as a patient with PTSD in the US. The last author is a white, cis-gender woman who has former experience as a HCW and family members with serious and persistent mental illness and addiction. Together, this research team has experiences with mental health care from the perspectives of patients, family members, and HCWs; these are all essential to name in this study that discusses mental health as a measure of wellbeing.

### 1.4. Purpose

This systematic review was conducted to rigorously identify and critically analyze the literature on moral injury and wellbeing (personal and professional) in healthcare settings. This review will contribute to future research by providing foundational knowledge on moral injury and its associated outcomes. The primary guiding research question for this review is:

What is the relationship between moral injury and wellbeing (personal and professional) amongst HCWs?

To best answer the research question, this review will include a summary of concepts, theories, methodologies, and results/findings found in prior studies of the association of moral injury and wellbeing.

## 2. Materials and Methods

### 2.1. PRISMA-P Protocol Overview

This systematic review is registered with the Open Science Framework (https://osf.io/q96bp/ accessed on 27 June 2021) and follows the PRISMA-P Systematic Review Protocol [35,36]. Guidelines from Boland and scholars and the Centre for Reviews and Dissemination were followed [37,38]. A mixed methods approach was used for this review including both qualitative and quantitative articles. Two reviewers (PT and AN) followed the same systematic search process to collect data based on the pre-established PRISMA-P protocol. A university-based research librarian supported the development of this protocol in the areas of information sources, inclusion/exclusion criteria, and search strategy.

### 2.2. Information Sources

The information sources for this review included healthcare-related and spirituality databases that included both peer-reviewed and gray literature. The following databases were searched: Academic Search Complete, ATLA, Dissertations & Theses Full Text, Google Scholar, Open Gray, Philosopher’s Index, PubMed, Religion Database, Social Services Abstract, SocIndex, and Web of Science. To frame the search strategy for this review, the SPICE framework was used to describe the parameters of the study. SPICE represents: the setting (S), which is healthcare settings; the population (P), which includes all HCWs; the interest (I), which is moral injury; comparison (C), though this review did not use a comparison group; and last, evaluation (E), which represents the association examined between wellbeing and the interest of moral injury [39].

### 2.3. Inclusion and Exclusion Criteria

The articles included in this review were selected if they met specific parameters focusing on the experience of moral injury in healthcare amongst healthcare workers. The review included both open access and fee-based articles accessible to the University of Denver Library. Dissertations and gray literature were included in the search because of the relatively recent emergence of this area of study. Regarding event history, studies before and during the COVID-19 pandemic were both included in this study. Articles were included if they went through a peer-review process and were empirically based. Differentially, studies that were related to moral injury were included in the study, and studies related to moral distress were excluded from the study. Commentary, conceptual, and review articles were excluded from this review. This study excluded articles that were not in English.

### 2.4. Search Strategy

The search was conducted using Boolean/phrase operators. The Boolean operators used were: (“moral injur*”) AND (wellbeing or well being or well-being or burnout or burned out or burnt out or compassion fatigue or retention, turnover, mental health, depress*, anxiet*, trauma, or stress) AND (healthcare professional or healthcare clinician or clinician or doctor or physician or nurse or nurse practitioner or physician assistant or social work* or clinical social work* or psychologist or psychiatrist) NOT (military, “active duty”, veteran, army, navy, “air force”).

Once the search protocol was implemented across all sources, the articles were imported into a reference management system, Zotero. All identified titles and abstracts were screened for the inclusion of moral injury and wellbeing outcomes within samples of HCWs. A title and abstract screening form was created to ensure that all articles were consistently screened using the same criteria. Once the articles were screened, conceptual articles, reviews, and letters to the editor were removed from the selected articles.

Prior to extracting data from the articles, PS emailed all the first authors of the included articles. This email requested their consultation, informed them that their article is going to be included in this systematic review, and inquired about any ongoing research about moral injury and wellbeing in healthcare.

### 2.5. Data Extraction

A data extraction form was created based on Cochrane’s data extraction template [40] and was used to extract data from the empirical studies included in this review. The same two reviewers, or henceforth “coders”, who systematically searched for the articles also extracted the data from the articles included in this study [40]. Using a data extraction tool, both coders reviewed and annotated each article in depth. Each coder was responsible for responding to each question in the data extraction tool for each independent article included in this review. After data extraction was completed individually by each coder, both coders met over a series of meetings to compare their data extraction responses. All discrepancies in the data extraction were reinvestigated by the two coders together until an agreement was reached.

### 2.6. Data Synthesis and Analysis

After the data were accurately extracted from both qualitative and quantitative papers, a standard review table was created to visually synthesize the included articles in this review. The table includes the following information: authors, publication year, location, aims, theoretical framing, sample (size, description, method), study design and methodology, outcome variables, study findings/results, and a conclusion statement about moral injury and wellbeing.

The analysis of this review included a synthesis of both qualitative and quantitative studies. The synthesis includes descriptive study data as well as conceptual and methodological summaries. The review table provides an accessible summary of the current state of research on moral injury and wellbeing, and the research is more deeply critiqued in the following narrative.

Quality Appraisal and Risk-for-Bias Assessment. The quality appraisal of the studies included in this review was conducted after data extraction to reduce any reviewer bias from the author while extracting data [37]. As per the Centre for Reviews and Dissemination’s and Boland and scholars’ guidance, [37,38], the Joanna Briggs Institute’s (JBI’s) critical appraisal and bias assessment tools were chosen [41,42,43]. The JBI’s quality appraisal tools were selected due to the validation of these tools in healthcare settings as well as the fit of the tools for various study designs.

From the JBI, three critical appraisal tools were chosen to align with the study designs used in the studies included in this review. Each tool asks a series of closed-ended questions regarding research questions, methodology, data collection, representation, interpretation, reflexivity, and ethics [41,42,43]. The response options for each question included, “yes”, “no”, “unclear”, and “not applicable”, and for the purposes of this review, we added a response option of “partially” to account for some of the studies that met partial criteria of the question. First, for the qualitative studies, the JBI Critical Appraisal Checklist for Qualitative Research tool was applied to the five qualitative studies (see Appendix A) [41]. For the quantitative studies, two different JBI critical appraisal tools were used to appraise the data of both cross-sectional and longitudinal study designs. The JBI Critical Appraisal Checklist for Cross-Sectional Studies tool was applied (see Appendix A) [42], and for the two longitudinal studies included in the review, the JBI Critical Appraisal Checklist for Case Series (Longitudinal) Studies was applied (see Appendix A) [43]. All of the studies generally met the quality appraisal checklist requirements, and a few studies are considered weaker studies due to lack of collection or reporting of demographic information and explicit naming and controlling for confounding variables in statistical analyses [16,18,19,44,45,46,47,48,49,50,51,52,53,54,55,56,57,58].

## 3. Results

The systematic search initially produced a total of 248 records that were identified from 12 databases, 0 articles from registries, and 2 articles from consulting experts in the field (see Figure 2). Duplicate articles were removed from the records (*n* = 104), leaving 146 records to be screened by title and abstract. After screening the 146 articles, 110 articles were excluded from the review for several reasons, including: keywords not being present in the title or abstract, no mention/measure of moral injury, no study of additional wellbeing measures, no association statistical analyses, and the articles were news articles, conceptual pieces, letters to the editor, or reviews. Finally, 36 articles were sought for retrieval, and 35 articles were successfully accessed. All 35 articles were assessed for eligibility, and 17 records were excluded for a variety of reasons, including: the article was a conceptual paper or review paper, had no measure of moral injury, used the wrong statistical analysis (not addressing the research question), or was a scale development paper (see Figure 2). The final group of articles retained in this review included 15 studies presented in 18 articles. Both quantitative (*n* = 13) and qualitative papers (*n* = 5) were included in order to understand both the occurrence and experience of moral injury, as this area of research is developing.

### 3.1. Sample Characteristics

Of the fifteen studies identified, two studies did not collect or disclose demographic data [19,44]. One study did not report demographic information despite disclosing that they collected it [45]. Only six articles included in the review collected information on the racial identity of their participants, and all of these articles used quantitative methodology [47,51,52,53,57,58]. The majority of participants were of White or Asian ethnicity. Many different types of HCWs were included in the empirical studies in this review. Most of the participants were doctors and nurses, but other healthcare workers, psychologists, directors, and non-clinical staff were included in smaller numbers within larger physician and nursing samples [16,18,45,48,50,51,52,53,55,56]. Generally, the mean age was found to be between 20 and 41 years old across all studies except one study, which categorized the mean age to be under 55 years old [53]. Age was measured in both categorical and continuous formats in these studies.

Some of the studies focused on new HCWs, either in training programs within their first few years, or medical students [19,45,46], while other studies had a minimum years of experience requirement [18], with many participants often having more years of experience [16,44,49,51,52,53,57,58]. Some studies did not have a requirement for the number of years of work experience [47,48,50,54,55,56]. Across the studies, other variables were collected such as marital status, education degree, and religious affiliation. Finally, seven of the articles were based in the United States, three in the United Kingdom, two in mainland China, one in Australia, three in Israel, one in Turkey, and one study that sampled from both Austria and Italy.

### 3.2. Synthesis of Moral Injury and Wellbeing

Current empirical evidence suggests that moral injury is correlated with wellbeing outcomes for healthcare clinicians. Findings demonstrate that moral injury is positively correlated with wellbeing outcomes [16]. The association between moral injury and wellbeing are summarized in the two major subcategories reflecting the organization of the current literature: personal wellbeing and professional wellbeing. An overview of the descriptive, conceptual, and methodological findings is found below (see Table 1).

Definition of Moral Injury. In this review, three papers use Shay’s definition [11], ten papers use Litz and Kerig’s definition, and three papers reference both definitions. One paper references a systematic review on moral injury as the citation for moral injury [45], and this review cites both Shay’s and Litz and Kerig’s definition as well [11,12,59]. Last, one team used a less common definition by Koenig, Ames, and Nash [60]. 

### 3.3. Qualitative Studies Summary

The five qualitative studies included in this review used a variety of approaches to understand the experience of moral injury amongst healthcare workers. Three of the papers used a phenomenological design, in which semi-structured interviews and thematic analysis were used to explore the experience of moral injury as well as its impact on their wellbeing [19,45,46]. All three of the studies using phenomenology studied samples of new or early career healthcare workers [19,45,46]. The other two qualitative designs used grounded theory [18] and a case study [44], and both these studies looked at experienced HCWs experiences of moral injury. Kreh et al. used both individual semi-structured interviews as well as focus groups [18]. Alexander used chaplain case notes over three years with a physician experiencing moral injury (over 45 notes) for the case study [44]. Three of the papers used convenience sampling to recruit their participants [18,19,44], while the remaining two studies used purposive sampling [45,46]. Thematic analysis was used for the grounded theory and phenomenological studies [18,19,45,46], and content analysis was used for the case study [44].

**Table 1 ijerph-20-06300-t001:** Systematic Review Summary Table.

Authors Year *Location*	Aims	Sample (Size, Description, and Method)	Methodology/ Design/ Theory	Concepts Studied (Variables)	Outcomes	Conclusion about Moral Injury and Wellbeing
**Alexander** **2020** [44] ***United States***	To offer an illustration of how moral injury interventions with veteran populations can inform care for physicians experiencing burnout.	*n* = 1 Female cardiologist with 20+ years clinical experience. Convenience Sampling	Qualitative Case Study Content Analysis No use of theory	Moral Injury (Shay’s Definition) Personal Wellbeing Compassion Fatigue Emotions Burnout	Themes: Use of clinical terms is not helpful in describing distress. Need to address the moral declination that impacts her personal wellbeing and work. Examination of all identities is essential. “Polarization” must be named in work vs. personal conflict.	Moral injury impacts personal wellbeing (adverse personal emotions, high stress, and polarization between work/personal life and beliefs) as well as professional wellbeing (burnout, compassion fatigue, and increased cynicism).
**Ball, Watsford, and Scholz** **2020** [45] ***Australia***	To analyze these data with regard to positive and harmful ways trainees have been impacted by their clinical work.	*n* = 14 Majority female (*n* = 11) psychologists in a medical center during the second year of their training program. Purposive Sampling	Qualitative Phenomenological Cross-sectional, semi-structured interviews Thematic Analysis [61] Recommended Biopsychosocial-spiritual model for theory.	Moral Injury (MI) [59] Vicarious Trauma (VT) Secondary Traumatic Stress (STS) Compassion Fatigue (CF) Burnout	Themes: Engagement with training and professional selves. Engagement with training and holistic selves. Self-Care	Trauma exposure could lead to STS, VT, and MI. MI can occur prior or alongside CF, and then burnout is a result of all these experiences.
**Benatov, Zerach, and Levi-Belz** **2022** [48] ***Israel***	To examine the moderating role of thwarted belongingness in the relationships between HCWs’ exposure to potentially morally injurious events (PMIEs) and moral injury symptoms, depression, and anxiety.	*n =* 296 Majority female, Israeli-born, and married. Mean age of 40.28 years, and included nurses, doctors, social and psychological care workers, and clinical support workers who mostly worked in hospitals. Convenience Sampling	Quantitative Cross-sectional Linger Regression Mediation-Moderation Modeling [62] Lietz’s framework of moral injury named in discussion [63].	Moral Injury *MISS-HP* Potentially Morally Injurious Events (PMIEs) *MIES* Anxiety *GAD-7* Depression *PHQ-9* Belongingness *Thwarted Belongingness (TB)*	Moral injury was positively correlated with anxiety, depression, PMIEs, and belongingness.	When healthcare workers are exposed to more PMIEs, they also experience moral injury symptoms, which is associated with anxiety and depression. The relation between PMIE and depression and anxiety is mediated via moral injury symptoms and moderated by thwarted belonging.
**Brown, Proudfoot, Mayat, and Finn** **2021** [46] ***United Kingdom***	To explore, “how do newly qualified doctors experience transition from medical school to practice” and “moral injury during transition”?	*n* = 7 New doctors (first 4 years) with an age range of 24–29 years, predominantly female, and who recently experienced a transition (<2 years). Purposive Sampling	Qualitative Hermeneutic Phenomenology [64]. Semi-Structured Interviews Thematic Analysis using an Interpretivist Paradigm [64,65,66]. Multiple and Multidimensional Transitions (MMT) Theory [67].	Moral Injury (Shay’s Definition) Transitional Experiences	Themes: The nature of transition to practice. The influence of community. The influence of personal beliefs and values. The impact of the unrealistic undergraduate experience.	Transition to practice was viewed negatively due to the lack of interpersonal support in 4-month rotations. Participants relied on the ethics of caring values to cope, but this in itself is troublesome and predisposes to moral injury.
**Chandrabhatla, Asgedom, Gaudiano, de Avila, Roach, Venkatesan, Weinstein, and Younossi** **2022** [49] ***United States***	To examine the relationship between burnout, second victim experiences, and moral injury experiences before and during the COVID-19 pandemic among hospitalists.	*n =* 81 Hospitalists between the ages of 20 and 40, with a stable partner/married, have children, and the majority of their work was clinical. Convenience Sampling	Quantitative Cross-sectional comparison Independent sample t-test No use of theory	Moral Injury *MIES* Burnout *Mini Z Burnout Survey* Second Victim Experiences *Second Victim Experience and Support Tool* Well-being *Flourishing Scale* *Satisfaction with Life Scale* Work Wellbeing *Work Wellbeing Scale*	Burnout levels reported were the same across pre COVID-19 and during COVID-19. An increase in reporting of second victim experiences during COVID-19, whether the hospitalist experiences burnout or not.	Moral injury was named as a predictive variable of burnout during COVID-19 in this study. During the pandemic, there was a higher rate of moral injury amongst burned out hospitalists.
**Dale, Cuffe, Sambuco, Guastello, Leon, Nunez, Bhullar, Allen, and Mathews** **2021** [47] ***United States***	This study investigated the occurrence that HCPs were experiencing MI, whether the experience of MI was related to co-occurring psychiatric symptomatology, self/others MI, and burnout.	*n =* 265 Majority white females with a mean age of 37.6 years old. Worked in a large city, have a college degree, and married or in a long-term relationship. Convenience Sampling	Quantitative Longitudinal Logistic Regressions Multilinear regression Multilevel modeling No use of theory	Moral Injury *MIES* Healthcare Morally Distressing Experiences *4-study related questions* Current Psychiatric Symptomatology *PHQ-9* *GAD-7* *PTSD Checklist-5* Workplace Burnout *Professional Fulfillment Index (PFI)*	Notably, longitudinally, self-moral injury was most impactful on experiences of burnout, and others moral injury was level influential on burnout. Higher levels of self-moral injury were correlated with higher levels of depression, anxiety, and PTSD, and other moral injury was only associated with depression.	When a healthcare worker conducts a moral injury themselves, they are most at risk for experiencing burnout. While witnessing others do things that healthcare workers find morally injurious can cause some depression, it is the individual moral injury that contributes to anxiety and PTSD.
**Kreh, Brancaleoni, Magalini, Chieffo, Flad, Ellebrecht, and Juen** **2021** [18] ***Austria and Italy***	To develop basic hypotheses regarding resilience and stress experiences of healthcare workers in the first phase of the COVID-19 pandemic.	*n =* 13 Healthcare workers (psychologists, physicians, and nurses) between the ages of 26 and 40, mostly female, with at least 5 or 10 years of experience for staff and clinicians, respectively. Convenience Sampling	Qualitative Grounded Theory Semi-Structured Interviews and Focus Groups Thematic Analysis [68]. No use of theory	Moral Injury (Shay’s Definition; Litz’s Definition) Psychological Safety Stress Institutional Support Resilience	Themes: Fear, guilt feelings, frustration, loss of trust, and exhaustion Casual factors: rapidly evolving situations with high uncertainty Stressors Resilience factors 3 developed hypotheses	Stress, power imbalance, and inability to separate home from work were all named as precursors to moral injury. Then, moral injury could result in poor mental health.
**Levi-Belz and Zerach** **2022** [50] ***Israel***	To highlight the emotional burden (depression and anxiety) among healthcare workers during COVID-19, and to further understand the direct and indirect role of PMIEs as well as the mediating role of stress and moral injury symptoms on depression and anxiety.	*n =* 296 Majority female, Israeli-born, and married. Mean age of 40.28 years, and included nurses, doctors, social and psychological care workers, and clinical support workers who mostly worked in hospitals, with an average 12 years of experience. Convenience Sampling	Quantitative Cross-sectional Pearson’s Correlations Structured Equation Modeling No use of theory	Moral Injury (Litz’s Definition) *MISS-HP* PMIE *MIES* Depression *PHQ-9* Anxiety *GAD-7* Perceived Stress *Perceived Stress Scale*	PMIEs were significantly positively correlated with depression and anxiety. Stress and MI were also found to be significant mediating variables between PMIE and anxiety and depression. The full model explained 63% variance in depression and 57% variance in anxiety.	This study highlights the relationship between moral injury and stress as well as moral injury increasing anxiety and depression.
**Murray, Krahé, and Goodsman** **2018** [19] ***United Kingdom***	To determine whether the concept of moral injury resonated with medical students working in emergency medicine and what might mitigate that injury for them.	*n* = 5 Medical students in prehospital care. Convenience Sampling —Sampled using critical case sampling	Qualitative Phenomenological Structured Interviews Focus Groups Thematic Analysis [61] No use of theory, names need for theoretical framing	Moral Injury (Shay’s Definition; Litz’s Definition) Trauma Exposure Social Support	Themes: What is Seen on Scene Material versus Human Resources The Complexity of Debrief	Moral injury acts as a reaction to witnessing trauma (but does not qualify as PTSD). Then, experiencing moral injury can lead to other wellbeing outcomes. Social supports and debriefing traumatic events are protective factors to reduce experiences of moral injury.
**Hines, Chin, Glick, and Wickwire,** **2021** [16] ***United States***	The purpose of the project was to quantify experiences of moral injury anddistress in HCWs during the first three months of the COVID-19 pandemic response.	*n* = 96 Majority female attending physicians with a mean age of 41 years old and an average of 14 years of experience in healthcare. Convenience Sampling	Quantitative Prospective Longitudinal Survey Design Descriptive Analysis Paired t-test Hierarchical Linear Modeling No use of theory	Moral Injury (Litz’s Definition) *MIES* Resilience *Resilience Scale* Distress *IES-R*	In the final model, stressful work environment was significantly associated with moral injury, while supportive work environment was nearly significantly associated with lower moral injury.	Stress and support are both related to moral injury, and stress was identified as a predictor to moral injury.
**Litam and Balkin** **2020** [51] ***United States***	To understand the extent to which healthcare workers experience moral injury while working in a pandemic.	*n =* 109 Majority white, female physicians and nurses, with an average age of 38 years old and an average of 12 years in healthcare. Convenience Sampling	Quantitative Cross-Sectional Survey Design Descriptive, correlational, and Multiple regression analyses No use of theory	Moral Injury (Litz’s Definition) *MIES* Professional Quality of Life *PROQOL:* -Compassion Satisfaction (CS) -Burnout -Secondary Traumatic Stress (STS)	STS was significantly associated with moral injury. Given the higher correlation between secondary traumatic stress and moral injury, a limited contribution of burnout was identified within the model, so burnout was removed.	STS was shown to significantly contribute to moral injury as a predictor. Burnout showed no association to moral injury, and CS was not significantly associated with moral injury.
**Mantri, Song, Lawson, Berger, and Koenig** **2021a** [52] ***United States***	To (a) characterize the changes in HP moral injury wrought by the pandemic over the course of 2020 and (b) identify potential predictors of moral injury amongst HPs.	*n =* 1831 Majority white, female, Christian, between the ages of 35–44, nurses and doctors, who are married. Snowball Sampling	Quantitative Cross-Sectional Survey Design Descriptive Student’s *t*-test Pearson’s Correlations Logistic Regression No use of theory	Moral Injury (Shay’s Definition) *MISS-HP* Religiosity *Duke University Religion Index (DUREL)* Burnout *Abbreviated MBI*	Results indicated that significant negative predictors of MISS-HP included ages of more than 55 years old, greater religiosity, direct experience with patients with COVID-19, divorced, and non-nursing professions.	Moral injury is a parallel construct to burnout. Moral injury has been suggested as a precursor to burnout [52], and it is possible that burnout rates will continue to increase as a lagging marker of ongoing moral strain. Personal identity factors impact moral injury.
**Mantri, Lawson, Zhizhong, and Koenig** **2021b** [53] ***United States***	To a) examine the prevalence of moral injury symptoms causing impairments in family, social, or occupational functioningand b) identify predictors of MI symptoms in bivariate and multivariate analyses.	*n =* 181 Majority white, male, physicians, with a majority of participants under the age of 55, who are Christian. Snowball Sampling	Quantitative Cross-Sectional Survey Design Descriptive ANOVA Student’s *t*-test Pearson Correlations No use of theory	Moral Injury (Shay’s Definition; Litz’s Definition) *MISS-HP* Clinical Characteristics Religious Characteristics *BIAC* Depression *PHQ-9* Anxiety *GAD-7* Burnout *MBI*	Moral injury symptoms were significantly more common among individuals who were more depressed, who were more anxious, or, especially, who indicated more burnout symptoms. In the final model, the strongest predictor of MI symptoms was burnout, followed by commission of medical errors in the past month, and religiosity at a trend level.	Moral injury is correlated with individuals with higher rates of depression, anxiety, and burnout. Committing medical errors, younger age, lower religiosity, and fewer years in practice were all significant predictors of moral injury. Moral injury mediates the relationship between experiencing transgressing moral code and the clinical outcomes.
**Morris, Webb, and Devlin** **2022** [54] ***United Kingdom***	To explore if healthcare providers in psychiatric settings are exposed to PMIEs, what the relationship between PMIEs and wellbeing are, and what the impact of COVID-19 is on PMIEs and wellbeing.	*n =* 237 Majority of participants were female, white British, between 21 and 30, and unregistered nurses. Convenience Sampling	Quantitative Cross-sectional Survey Design Spearman Rank-Order Correlations Bootstrapped Regressions No use of theory	Moral Injury/PMIEs (Litz’s Definition) *MIES* Wellbeing (Subscales: Burnout, Secondary Trauma, Compassion Satisfaction) *ProQOL-5*	Moral injury has significant positive associations with burnout, secondary traumatic stress, and significant negative associations with compassion satisfaction.	Moral injury was predictive of higher secondary trauma and burnout as well as lower self-compassion amongst healthcare workers.
**Ulusoy and Çelik** **2021** [55] ***Turkey***	To determine burnout levels and possible related psychological processes such as psychological flexibility, moral injury, and values among healthcare workers after the first year of the COVID-19 pandemic.	*n* = 124 The sample was majority female doctors with a mean age of 33.3 years old. Convenience Sampling	Quantitative Cross-sectional Survey Design Correlation Analysis Multiple Linear Regression No use of theory	Moral Injury (Litz’s Definition) *MIES* Psychological Flexibility *Acceptance and Action Questionnaire-II* Burnout *MBI* Depression and Anxiety *Depression Anxiety Stress Scale 21* Values *Valuing Questionnaire*	Depression and anxiety were the only significant predictors of emotional exhaustion. Moral injury was the only significant predictor of depersonalization. Moral injury, days worked during COVID-19, and value obstruction were the significant predictors for personal accomplishment.	This study demonstrates associations between moral injury and burnout, specifically moral injury as a predictor of depersonalization and personal accomplishment within burnout.
**Zerach and Levi-Belz** **2021** [56] ***Israel***	The objectives of this study were to examine patterns of exposure to potentially morally injurious events (PMIEs) among HSCWs and their associations with MI, mental health outcomes and psychological correlates.	*n =* 296 Majority female, Israeli-born, and married, with a mean age of 40.28 years, and included nurses, doctors, social and psychological care workers, and clinical support workers who mostly worked in hospitals. Convenience Sampling	Quantitative Cross-sectional survey Design Latent Class Analysis No use of theory	Moral Injury *MISS-HP* Potentially Morally Injurious Events (PMIEs) *MIES* Depression *PHQ-9* Self-Criticism *Factor from Depressive Experiences Questionnaire* Trauma *International Trauma Questionnaire for PTSD and C-PTSD* Self-Compassion *Self-Compassion Scale—Short Form*	Participants who had high exposure or betrayal exposure to moral injury experienced more PTSD and moral injury symptoms than those with minimal exposure. Those in the high exposure group also had more depressive symptoms. Additionally, those in the high exposure and betrayal only exposure groups had higher rates of self-criticism and lower self-compassion.	This study highlighted the relationship between moral injury and trauma (PTSD), mental health (depression), self-criticism and low self-compassion.
**Zhizhong, Al Zaben, Koenig, and Ding** **2021** [57] ***China***	To examine the relationship between spirituality, moral injury, and mental health among physicians and nursesin mainland China during the COVID-19 pandemic.	*n =* 3006 Majority Han, female, doctors, with bachelor’s degree, married, and not affiliated with religion, with an average age of 35 years old, with an average of 12 years of practice. Snowball Sampling	Quantitative Cross-Sectional Survey Design Descriptive Pearson’s Correlations Students *t*-test ANOVA Hierarchical Linear Modeling No use of theory	Moral Injury (Litz’s Definition) *MISS-HP* Spirituality Depression *PHQ-9* Anxiety *GAD-7*	Spirituality was positively correlated with moral injury, depressive symptoms, and anxiety symptoms) after controlling sociodemographic variables.	Moral injury is correlated with mental illness. Those with higher spirituality were associated with experiencing higher moral injury. Moral injury *was* a mediating variable but *was not* a moderating variable between spirituality and depression/anxiety.
**Zhizhong, Koenig, Yan, Jing, Mu, Hongyu, and Guangtian** **2020** [58] ***China***	To assess the psychometric properties of the 10-item Moral Injury Symptoms Scale-Health Professional (MISS-HP) among healthcare professionals in China.	*n =* 3006 Majority Han, female, doctors, with bachelor’s degree, married, and not affiliated with religion, with an average age of 35 years old, with an average of 12 years of practice. Snowball Sampling	Quantitative Cross-Sectional Survey Design Pearson’s Correlations Students *t*-test ANOVA No use of theory	Moral Injury (Litz’s Definition) *MISS-HP* Spirituality Depression *PHQ-9* Anxiety *GAD-7* Well-being *Secure Flourish Index (SFI)* Burnout *MBI-HSMP*	Moral injury had a small significant inverse correlation with personal accomplishment and a significant moderate inverse association with SFI. Otherwise, moral injury had a significant moderate positive association with the remaining constructs: PHQ-9, GAD-7, emotional exhaustion, and depersonalization.	Moral injury is found in increasingly stressed healthcare professionals, and moral injury is correlated with depression, anxiety, burnout (all three subconstructs), and flourishing.

### 3.4. Quantitative Studies Summary

Eleven studies used cross-sectional study design [48,49,50,51,52,53,54,55,56,57,58]. One study used a prospective longitudinal survey design [16], and another study used a case series longitudinal design [47]. Nine of the studies [16,47,48,49,50,51,54,55,56] used convenience sampling methods to recruit their participants and the remaining four quantitative studies used snowball sampling methods to recruit their participants [52,53,57,58]. Across the quantitative studies, the Moral Injury Events Scale (MIES) was used in six papers to measure moral injury [69], and the remaining four papers [52,53,57,58] used the Moral Injury Symptom Scale–Healthcare Provider (MISS-HP) [70], which was adapted from the Moral Injury Symptom Scale Military Short Form (MISS-M-SF) [60]. Three papers used both the MIES and the MISS-HP [48,50,56]. The MIES scale is a more generalized scale to measure moral injury, while the MISS-HP is a healthcare setting specific scale.

The quantitative studies used various measures and constructs to measure the concept of wellbeing amongst HCWs, accounting for both professional wellbeing outcomes and personal wellbeing outcomes [16]. The most common constructs that were analyzed in the quantitative articles were burnout, compassion fatigue/satisfaction, mental health, and spirituality/religiosity. For burnout, almost all of the tools used included an iteration of the Maslach Burnout Inventory (MBI) [53,55,57]; one study used the abbreviated MBI (aMBI) [52], another used the MBI–Human Services Survey for Medical Professionals (MBI-HSMP) [58]. When studying burnout, a few studies did not use an iteration of the MBI. Litam and Balkin [51], as well as Morris and scholars, instead used the Professional Quality of Life (ProQOL) with a subscale of burnout [54], while Chandrabhatla and scholars used the Mini Z burnout survey [49], and the Professional Fulfillment Index was also used [47]. For compassion fatigue/satisfaction, Litam and Balkin and Morris and scholars also analyzed this construct using the ProQOL scale with a subscale of compassion satisfaction [51,54]. The other studies that analyzed compassion fatigue/satisfaction were qualitative. A variety of scales were used to measure spirituality/religiosity. All of the scales used were validated and reliable. The scales used in the studies to measure spirituality/religiosity included the Duke University Religion Index (DUREL) [52], the Belief into Action Scale (BIAC) [53], and visual analogue scales [57,58]. Last, mental health was measured through secondary traumatic stress, depression, and anxiety, using the ProQOL [51], PHQ-9, GAD-7 [47,48,50,53,56,57,58], the Global Mental Health–K6 Scale [56], and through the Depression Anxiety Stress Scale [55].

### 3.5. Personal Wellbeing

The following constructs were used to measure personal wellbeing: “personal wellbeing” [44,58], emotions [44], transitional experiences [46], stress/distress [16,18,47,50,56], resilience [16,18,47], spirituality/religiosity [52,53,57,58], psychological safety [18], social support [19], thwarted belonging [48], flourishing [49], life satisfaction [49], psychological flexibility [55], self-criticism [56], self-compassion [56], valuing [55], and mental illness, including both depression and anxiety [48,50,53,55,57,58].

All studies found an association between moral injury and personal wellbeing. One qualitative study found that moral injury impacted personal wellbeing, specifically increasing stress, emotions, and polarization between personal and work life [44]. Other studies found that adverse personal wellbeing is a risk factor for experiencing moral injury [16,46,53]. For example, Brown and scholars [46] found that the nature of transitional experiences can cause disruptions in physician wellbeing, which could then lead to moral injury. Other studies named moral injury as a mediating variable between multiple personal wellbeing outcomes [18,19,48,50,57,58]; such as, when a healthcare worker experiences stress, they can then experience moral injury, which could lead to adverse mental health outcomes [18,48,50]. Interestingly, resilience was not associated with moral injury across these studies [16,18,47]; yet, Zerach and Levi-Belz demonstrated a relationship between moral injury and self-criticism and low self-compassion [56].

### 3.6. Professional Wellbeing

The studies that measured professional wellbeing included these constructs: compassion fatigue/satisfaction [44,45,51]; trauma exposure [19,47]; vicarious trauma [45]; secondary traumatic stress [45,49,51,54]; burnout [44,45,47,49,51,52,53,54,55,58]; and institutional support [18]. Burnout was found to be both associated [19,45,52,53,58] and not associated with moral injury [51]. Of those studies that found that moral injury and burnout are associated, some studies described burnout as an outcome of experiencing moral injury [19,45,53,58]; alternatively, one described burnout as a parallel construct to moral injury, where the constructs impact one another, but they co-exist [52], and another two studies found burnout to be a predictor of moral injury [49,55]. In three of the studies included in this review, compassion fatigue was found to be an outcome of moral injury [44,45,54], and in another study, was found not to be significantly related to moral injury [51]. 

Trauma exposure, across the studies included in this review, was measured using three different types of trauma (trauma exposure, vicarious trauma, and secondary traumatic stress). All three forms of trauma were framed as predictors of moral injury [45,51]. Multiple studies that identified trauma, or more specifically a trauma-response, as an outcome of moral injury named the clinical diagnosis or PTSD, which does not always occur after experiencing moral injury [19,47,53,56]. Institutional support was named as a protective factor for moral injury, and power imbalances were identified as a risk factor for experiencing moral injury [18]. 

### 3.7. Use of Theory

Out of the 18 articles included in this study, one paper included a specified theory [46]. The qualitative study that used this theory aimed to answer a question regarding how newly qualified doctors experience the transition from medical school to practice [46]. The authors of this paper suggested the use of Multiple and Multidimensional Transitions (MMT) Theory [67]. The theory in this study was used to contextualize the transitional time in which the participants could have higher levels of vulnerability rather than directly using theory to frame moral injury [46]. Two additional articles acknowledged the need for theoretical framing when studying moral injury amongst HCWs [19,45]. Ball and scholars [45] named the biopsychosocial–spiritual model [71] as a potential theory to contextualize moral injury in healthcare. Murray and scholars acknowledged the need for theoretical framing, but they did not recommend any specific theories or frameworks [19].

## 4. Discussion

This systematic review identified 18 original empirical articles that examined the relationship between moral injury and personal and professional wellbeing amongst HCWs. This review found that there is a direct link in the literature between moral injury and wellbeing amongst HCWs. An association between moral injury and wellbeing was identified across both personal and professional wellbeing, but the temporal order of moral injury and wellbeing-related outcomes remains unclear. Across these studies, there were relationships found between moral injury and related constructs including burnout, trauma (vicarious trauma and secondary traumatic stress), compassion fatigue, mental health (depression, anxiety, and PTSD), and stress. Since this review identifies associations between moral injury and several wellbeing outcomes, it is demonstrating the vast impact that the experience of moral injury has on HCWs.

### 4.1. Theoretical Framing

As found in this review, there was a lack of theoretical orientation in the studies, and this reflects the early stage of moral injury research in healthcare. The studies identified in this review are essential in contributing to the theoretical framing of moral injury in healthcare as they provide information on the phenomenon of moral injury and its association to other related constructs of wellbeing. However, these studies do not conceptually illuminate how personal and professional identities contribute to the experience of moral injury. Even further, a commonly named risk factor for moral injury has been identified as power imbalance (systemically and interpersonally); yet, power was not measured across any of the studies in this review. This issue could be addressed by using existing theoretical frames to articulate the role of power/imbalance in the experience of moral injury. Specifically, using systems-level theory aligns with the concept of moral injury, as moral injury speaks to systems-level causes and solutions as opposed to burnout, which generally alludes to individual level causes, symptoms, and solutions [72,73,74].

### 4.2. Power as a Measured Construct

In the study of moral injury in healthcare, there remain numerous gaps in understanding the true nature of moral injury and how to intervene. First, in the study of moral injury, power is an essential asset to consider. Often, moral injury occurs when a power imbalance occurs between two or more people [11]. For example, if a supervisor requires a staff member to complete a task that the staff member disagrees with, the supervisor has the power to enforce the task completion, and the staff can in turn experience moral injury.

### 4.3. Consequences of Moral Injury for Healthcare Workers

The experience of moral injury captures a nuanced response to the challenges that HCWs face daily. HCWs, specifically doctors and nurses included in this review, work with people experiencing complex health demands, and the solution-searching in the midst of crisis that is required of HCWs is demanding. When a HCW experiences a moral injury, they can experience a variety of known and unknown outcomes. Studies included in this review demonstrate that experiencing moral injury causes emotional consequences of guilt and shame as well as values consequences of internal confusion with oneself. A HCW experiencing guilt and shame surrounding their work while caring for patients is having a juxtaposed experience. It is challenging for HCWs to be in the spotlight of implementing healthy practices and policies when they themselves are feeling upset or confused about the decisions they have made or witnessed within healthcare systems. These consequences of moral injury could contribute to HCWs feeling lonely, retreating from social and institutional support, and in turn leaving healthcare. Yet, within the current study of moral injury, researchers have found that naming moral injury as an experience that healthcare workers may face reduces a sense of isolation, feelings of guilt and shame [70,75].

### 4.4. Contextualizing Moral Injury within Wellbeing

Essential components contributing to the greater context of HCW moral injury and wellbeing are pay inequity, high caseloads, crisis standard of care, staffing shortages, HCW abuse (from the system and the patients), and the privatization and profiting of healthcare services in capitalistic countries. It is important to note that with the rapidly emerging research on moral injury in healthcare, there are still many unknown consequences of moral injury. This review serves as a beginning insight into opportunities for further understanding the consequences of moral injury within pre-existing wellbeing indices. Because the main constructs found to be associated with moral injury were burnout, trauma, mental health, and spirituality/religiosity, each of these will be discussed in turn.

### 4.5. Burnout

Previous literature has continuously highlighted the experience of burnout as the predominant way in which professional wellbeing was measured in healthcare settings [72,76]. Professionally, wellbeing literature often debates the constructs of burnout and moral injury, and scholars either distinctly separate these two terms or use them interchangeably [77]. This review demonstrated no collinearity between the constructs, although they are indeed associated, thus confirming the differences between burnout and moral injury. Conceptually, the emotional consequences of moral injury align with the domain of emotional exhaustion within burnout [55], identifying a potential pathway of the relationship between moral injury and burnout. Another sub-construct of burnout is depersonalization, meaning feeling unlike oneself, and this is conceptually related to the consequence of internal confusion about oneself from moral injury.

### 4.6. Trauma

Trauma via multiple mechanisms, including secondary/vicarious trauma and primary trauma exposure, were found to be associated with moral injury, demonstrating the relevance of trauma in the experience of moral injury. Moral injury, as a phenomenon, is specific to high stakes situations. Healthcare, innately, is a high stakes environment, and often HCWs are exposed to high levels of trauma compared to the general public [54].

### 4.7. Mental Health

Moral injury and mental health demonstrated a consistent association across articles included in this review. When referring to mental health, this includes all diagnosable mental health disorders, but often arose as depression, anxiety, and trauma responses. In a recent scoping review of moral injury, a scholar listed primary and secondary consequences of moral injury, and they state that depression, anxiety, and self-harm are all potential symptoms caused by moral injury [75]. With the high rates of mental health diagnoses amongst HCWs in general [78], the current review contextualizes moral injury’s role alongside mental health. Specifically, this review did not measure for PTSD or acute stress disorder (ASD). While all moral injury experiences may not lead to PTSD or ASD, it is important to account for mental health diagnoses pertinent to trauma as trauma was repeatedly associated with moral injury across the articles in the current study.

### 4.8. Spirituality/Religiosity

Spirituality/religiosity demonstrated a strong association with moral injury across articles. Often individuals develop their moral orientation from their environment and systems they belong to, and historically, morals are often taught in systems of religion and spirituality (i.e., temples or churches). This association provides some insight into how some individual beliefs influence their experiences of moral injury. For example, what may feel right or wrong to one person may differ from the next person based on their belief system. Using spirituality/religiosity is one mechanism of measuring individual belief systems.

### 4.9. Measuring Moral Injury

This review also demonstrates the exploratory nature of studying moral injury in healthcare settings through sampling and data collection methods. Across qualitative and quantitative methods, convenience sampling was predominantly used. Qualitatively, the research questions focused on understanding the phenomenon of moral injury amongst HCWs. Quantitatively, most of the studies used cross-sectional data collection methods that supported a general understanding of moral injury through surveys and scales. The study of moral injury amongst HCWs is a concept that is in a foundational research stage.

### 4.10. Synthesis of the Literature

The articles included in this review explored the relationship between moral injury and wellbeing amongst healthcare workers around the globe. Both qualitative and quantitative articles were included in this review, and each methodology provided unique insights. Specifically, the quantitative articles presented rates and severity of moral injury and additional wellbeing outcomes, highlighting the significant relationships between moral injury and several additional measures of wellbeing. The statistical representation of this association is valuable; yet, the qualitative studies generally provided more complex, unique, and deep understanding on moral injury and wellbeing. The qualitative studies were able to identify causes, potential mediators between moral injury and wellbeing (e.g., material versus human resources [19]), complex emotional consequences of moral injury (e.g., fear and frustration [18]), and highlight imaginative solutions (e.g., debriefing sessions [19]) to ameliorate moral injury. Additionally, the qualitative studies’ samples differed from the quantitative samples, as the qualitative samples explicitly included or targeted students and trainees in their studies, whereas the quantitative studies did not explicitly name the inclusion of trainees in their samples.

Furthermore, a few differences were noted when comparing U.S.-based studies to studies from other countries. First, similar sample differences were noted when comparing the United States to studies from other countries in this review as noted when comparing the qualitative to quantitative studies. The U.S.-based studies did not explicitly include students and trainees, where there was more inclusion of these groups around the world. Moreso, the samples in the United States predominantly included physicians and nurses, whereas non-U.S.-based studies included a wider range of health professionals, including psychologists, social and psychological care workers, and clinical support workers, acknowledging the need to support the entire healthcare workforce. Last, it is important to note that non-U.S.-based studies included more psychosocial measures of wellbeing beyond the traditional understanding of professional and personal wellbeing. Other countries included measures of belonging, social support, psychological safety, psychological flexibility, and values-based questions, which add to the depth of understanding of moral injury and wellbeing amongst healthcare workers. In all, while there are differences across countries, samples, and methodologies found in this review, this work speaks as a cohesive body, offering a significant insight into the impact moral injury has on HCWs’ wellbeing.

### 4.11. Future Directions in Research

In forefronting power dynamics in the study of moral injury, scholars can be more inclusive of who is represented in their study samples, recognizing that other healthcare staff (i.e., social workers, housekeeping staff, certified nursing assistants) are at risk for experiencing moral injury due to their lack of power within the healthcare system. Next research steps should include the study of social workers and mental health workers in the understanding of moral injury in healthcare. Specifically, the phenomenon of moral injury should be better understood from the ground up as it uniquely impacts professions differently. Future moral injury exploratory research should also include the measure of power/imbalance, as empirical support on the role of power in the experience of moral injury is needed. When studying power/imbalance, the impact of social support and collaboration on moral injury and wellbeing should be examined.

In shifting from exploration to intervention research, much work is to be completed. Current interventions on moral injury are individual interventions; however, when power is named as a tenet or assumption of moral injury, aligned interventions could, and should, be directed at the systemic level. Addressing moral injury at the systemic level is essential, and moral injury healthcare research has not studied the outcomes of systems-level interventions. Current literature has tested a few interventions for healing the experience of moral injury, such as using acceptance and commitment therapy and hosting moral rounds or lunches at work [79,80].

## 5. Limitations

This review is not without limitations. Moral injury, which is not always named as “moral injury”, is a concept that is reported in articles beyond the ones included in this review under varying additional terminology (e.g., moral distress, moral emotion, and moral wrongdoing). For example, articles that used the term moral distress were excluded from this review, and while that provided a specific scope of research when studying the association of moral injury and wellbeing, moral distress and moral injury are terms that are often conflated in the literature, and studies may have been excluded that would otherwise fit the scope of this research. Moreso, moral injury is a concept which is recently gaining traction, and there are multiple definitions used for this term. Moral injury in this review was not limited to one definition of the term [11,12], and within the literature reviewed, multiple different scales to measure moral injury were used, making comparisons of rates of moral injury and associations to other wellbeing outcomes less consistently reliable. Additionally, this review used broad inclusion criteria for the term wellbeing, including wellbeing across professional and personal domains. While broad inclusion criteria on wellbeing were essential due to the emerging nature of moral injury in healthcare research, it also may have provided too many associations of moral injury and wellbeing to consider. Further, although a strength of this review was that two coders followed the review protocol, selection bias may have still occurred when selecting articles due to our positionalities. Last, the generalizability of this review is limited as several of the articles included in this review are based in the United States. The United States’ healthcare system fundamentally operates and is valued differently than in the majority of the world. Specifically, the capitalistic frame of healthcare services in the US, the US policy implemented for public health crises, and the lack of respect for the healthcare workforce all contribute to wellbeing in uniquely distinct ways than in other countries.

## 6. Conclusions

These studies, and their associated methodologies, each contribute to the greater meaning of moral injury in healthcare. This review begins to transition research from exploration and understanding to association and trends, in synthesizing the connection of moral injury to other wellbeing constructs in the field of healthcare. In the context of moral injury, its association with trauma is unsurprising; yet, a deeper understanding of the nature of the relationship between trauma and moral injury is urgently necessary. Future research should seek to incorporate additional tools that measure individuals’ belief systems in the study of moral injury to gain a better understanding of how non-personal beliefs are associated with moral injury. Additionally, future research should study moral injury across the allied health professions. This review makes the first steps in this identification process and in addressing gaps in the existing interventions, and begins to thread together rates, narratives, and conceptual framing connecting moral injury, burnout, and mental health outcomes. Future research should include intervention research to help identify strategies to ameliorate the experiences of moral injury and its associated outcomes.

## Figures and Tables

**Figure 1 ijerph-20-06300-f001:**
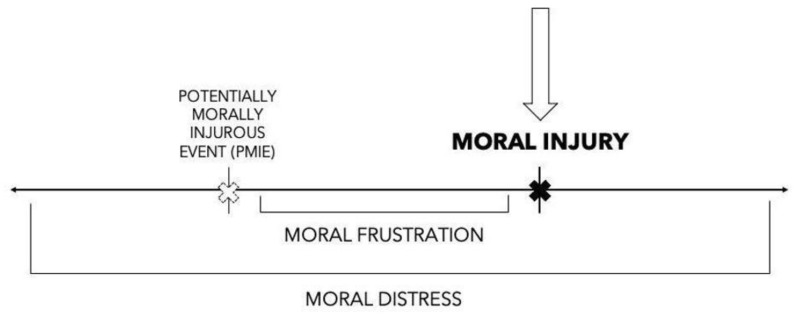
Continuum of Moral Experiences.

**Figure 2 ijerph-20-06300-f002:**
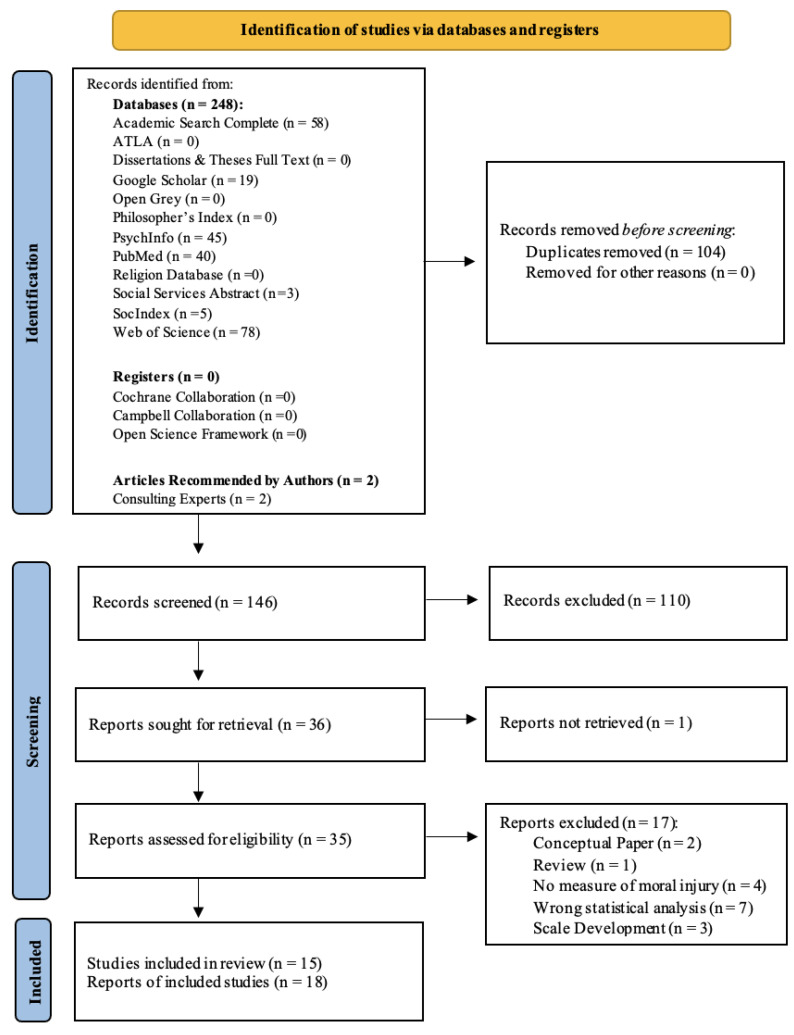
PRISMA-P Consort Diagram.

## Data Availability

Data sharing not applicable.

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
