# Peer review of "The Association of Moral Injury and Healthcare Clinicians’ Wellbeing: A Systematic Review"

_ijerph, 2023, doi:10.3390/ijerph20136300_

Round 1

Reviewer 1 Report

Dear Authors,

I have reviewed the manuscript. Although the manuscript reads well however, please pay attention to the following points to improve the clarity of the paper overall. 

1. Please arrange the table 1 and 2 to provide more clarity. As of now, the readers will have difficulty to understand the key message. If this table does not fit in the text then the best way would be to put it in the supplement and mention the result. The readers are interested in the actually quality-rather the intricacy of the table. We can always look the supplemental for details. The table can be summarized in one paragraph and mentioned in the supplemental. 

2. Figure 2, the PRISMA chart is not placed properly and not clear. Please use one page only for this image. 

3. Table 4. Systematic Review Summary Table: Although I understand what the authors are trying to say using this table however, please the table in such a way that there is a clear representation of the data. As of now, it's difficult to find the text that belongs to the correct rows and column. Reading the table is very tiring. In addition, merge a few column to reduce the word count. The words form one row is mixing with the other row. 

4. Fonts are different for references.

I wish you all good luck. Thank you.

Author Response

Dear Reviewer,

Thank you for providing your time and a review for our article.

Please see the attachment for our responses and revisions. 

Pari

Reviewer 2 Report

In this manuscript, the authors performed a systematic review including 18 qualitative and quantitative empirical articles that examined the relationship between moral injury and personal and professional wellbeing amongst HCWs, and found that there is a direct link between moral injury and wellbeing amongst HCWs. This study topic is relatively new and the content is rich, which is a good review. I have only a minor point: In the RISMA-P Consort Diagram, Records screened should be 144?

Author Response

(The authors gave the same response as above.)

Reviewer 3 Report

This is an important paper discussing moral injury in healthcare. As a systematic review, it is relatively small-n, with only 18 qual and quant papers total. This question of operationalizing moral injury in the context of healthcare worker wellbeing is an important one, and this is an important contribution.

The question also overlaps with the overall trend (especially in the US) for patients to be abusive towards healthcare providers, during COVID-19 but also generally. More could be said in the paper of this overlap.

There should be more said about the lack of generalizability of these US findings for other parts of the world, which are less barbaric, and continue to have more respect for medical care workers. Although, certainly, the questions concerning burnout are likely more transferable. There also could be made more discussion of how the low wages, increased need, and profiteering of the medical industry participates in the social determinants of mental health for healthcare workers abused by patients. Overwork and lack of safe spaces, and a general grinding down of goodwill are all part of the equation.

The PRISMA diagram was cut-off in the version I received to review, so I cannot fully review that portion of the paper. Also unclear whether this systematic review is already registered or not. Also unclear whether the systematic review is restricted to papers documenting US cases, or whether they are broader. 18 papers for worldwide would be too thin to generate a meaningful systematic review.

The authors could emphasize that moral injury is not normally glossed as moral injury, or certainly not conveyed in those terms. So insofar as the article wishes to track actual behavior and experience rather than language is important. This distinction is important, and of course, this is a limitation of systematic reviews of this sort (which are large database analyses based on keywords). So, perhaps something should be said here, as it is likely that the majority of papers that actually are noting some sort or form of moral injury do not use any of the gathered keywords. This is a fundamental philosophy of science question/problem that too often is recapitulated, and creates linguistic analyses without actually having a full picture of the relevant possible data.

Author Response

(The authors gave the same response as above.)

Reviewer 4 Report

Dear Authors,

I really appreciate your paper because of the relevance of the content. In order to improve, please find my comments below:

1. Better clarification of the criteria you adopted to select the 18 papers out of 240. is it related to the content, the presence of quantitative methodology? Is it possible to have a second cluster of papers between 240 and 18? for example some dozens of papers that have only qualitative approach or other criteria?

2. Actually it will be useful to dedicate some raws to clarify the relations between personal wellbeing, professional wellbeing and burnout. In you framework is burnout an antecedent or a consequence of wellbeing? Professional wellbeing consist only of depression, overloaded stress. Is professional wellbeing a dimension of organizational wellbeing or is completely different in your framework?

3. In some part of the paper you mentioned mental health, what do you mean really? 

4. You do not mentioned collaboration that could impact on wellbeing.

5. You have a detailed description of the paper you analyzed but i do not see any interpretation. For example, a comment on difference between qualitative and quantitative papers. Do you see country specific aspects? Can you underline differences among different professional groups, nurses, psychologist ecc?

Author Response

(The authors gave the same response as above.)
